# Lyapunov Matrix-Based Guaranteed Cost Dynamic Positioning Control for Unmanned Marine Vehicles With Time Delay

1st Xin Yang
*College of Navigation*
*Dalian Maritime University*
Dalian, China
yangxin3541@163.com

2nd Li-Ying Hao*
*College of*
*Marine Electrical Engineering*
*Dalian Maritime University*
Dalian, China
haoliying_0305@163.com

3rd Tieshan Li*
*College of Automation Engineering*
*University of Electronic Science*
*and Technology of China*
Chengdu, China
tieshanli@126.com

4th Yang Xiao
*Department of Computer Science*
*The University of Alabama*
Tuscaloosa, USA
yangxiao@ieee.org

5th Guoyong Liu
*College of*
*Marine Electrical Engineering*
*Dalian Maritime University*
Dalian, China
liuguoyong0806@163.com

*Abstract*—This paper presents a Lyapunov matrix-based guaranteed cost dynamic positioning controller for unmanned marine vehicles (UMVs) with time delays. A novel Lyapunov–Krasovskii functional (LKF) is introduced, which enhances the analysis of time delays and system states. The controller design leverages the LMI framework alongside Jensen's inequality to determine sufficient criteria for its feasibility, ensuring that the UMVs' state errors gradually reduce to zero and providing an adaptive $H_\infty$ performance guarantee. Additionally, the cost function is upper-bounded, and the effectiveness of the method is demonstrated through simulation results.

*Index Terms*—Lyapunov matrix, time delays, guaranteed cost control (GCC), dynamic positioning (DP), unmanned marine vehicles (UMVs)

## I. INTRODUCTION

Unmanned Marine Vehicles (UMVs) play a pivotal role in enhancing maritime safety and security by performing high-risk operations effectively without compromising human lives, thereby revolutionizing search and rescue missions and coastal surveillance [1]–[3]. Compared to traditional anchor mooring, dynamic positioning (DP) offers a more versatile, precise, and environmentally friendly method for positioning vessels, making it particularly suitable for use in complex or dynamic marine environments [4]. Over the years, numerous control strategies have been proposed to ensure robust DP control in UMVs. For instance, [5] introduces a dynamic output feedback control method, specifically tailored for DP ships to counter denial of service attacks. In [6], the design of an adaptive sliding mode fault-tolerant compensation mechanism is presented, targeting the maintenance of DP control in UMVs despite thruster faults and unknown ocean disturbances. It is crucial to recognize that time delays are typically inevitable [7]–[9]. Consequently, there is an urgent need to develop a strategy to compensate for these time delays.

In DP systems for UMVs, time delays due to network-mediated signal and control command transmission represents a significant challenge that often compromises system stability and performance [10], [11]. This issue has led to the development of various advanced time delays compensation methods [12]–[14]. Among these methods, enhanced time delays compensation approaches for autonomous underwater vehicles have shown promise [12]. In [13], model-free proportional-derivative controllers are innovatively incorporated into the Lyapunov–Krasovskii functional (LKF) framework to effectively counteract the impacts of delays. Advanced strategies utilizing Lyapunov matrix-based LKF methods have proven particularly effective. These approaches leverage comprehensive information about time delays and system states, providing control strategy that efficiently accommodates time delays systems. The primary motivation of this paper is to develop a complete LKF based on the Lyapunov matrix to mitigate the effects of time delays on UMVs.

On another research front, guaranteed cost control (GCC) has been extensively studied [15]–[17]. This strategy offers the advantage of setting an upper limit on a specified performance index, ensuring that any system performance degradation

This work was supported by the National Natural Science Foundation of China (Grant Nos: 51939001, 52171292, 61976033); Dalian Outstanding Young Talents Program(2022RJ05)

* Corresponding authors. Emails: haoliying_0305@163.com;tieshanli@126.com

remains below this predefined cost threshold. As vessels often navigate in complex and varied ocean environments, the impact of wind and wave disturbances becomes significant [17]. In response, [18] investigated a robust $H_\infty$ guaranteed cost controller aimed at enhancing path-following performance. The GCC method presented in [19] offers a way to reduce energy consumption for surface vessels in DP, thereby increasing its practical applicability. These results have inspired our research into GCC theory, particularly its application to DP ships. Thus, how to propose a guaranteed cost controller based on the Lyapunov matrix to achieve effective DP control for UMVs is the second research motivation of this paper.

The primary objective of this paper is to design a Lyapunov matrix-based guaranteed cost dynamic positioning controller, utilizing the LMI method to ensure stability. The paper's main contributions are evaluated in comparison to recent advancements in the field.

1) We propose a novel time delays compensation method for UMVs that incorporates more detailed time delays and state information by employing a Lyapunov matrix-based complete-type LKF, which reduces conservatism compared to conventional time delays compensation techniques.

2) A novel guaranteed cost DP control strategy is designed, which ensuring the stability of DP systems for UMVs while providing an upper bound on a prespecified cost function.

The remainder of this paper is structured as follows: Section II describes the UMVs model with time delays. Section 3 reviews basic concepts and preliminary results, which serve as the theoretical basis for the proposed LKF method based on the Lyapunov matrix. A complete-type LKF based on the Lyapunov matrix is presented in Section 4. Section 5 introduces guaranteed cost dynamic positioning controller. Finally, Section 6 presents simulations to illustrate the validity of the theoretical results.

## II. UMVs MODELING AND PROBLEM DESCRIPTION

### A. Dynamic modeling for UMVs

The UMVs model typically employs a three degrees of freedom motion equation to describe its dynamic behavior in the marine environment. These three degrees of freedom include yaw, surge, and sway. Therefore, the dynamic equations of the UMVs are often simplified and expressed in the following form [20]:

$$\xi \dot{v}(t) + \mathcal{C}v(t) + \mathcal{D}\lambda(t) = \mathcal{G}u(t), \quad (1)$$

$$\dot{\lambda}(t) = \mathcal{S}(\theta(t))v(t), \quad (2)$$

where matrix $\xi$ represents the inertia matrix, and the velocity vector $v(t) = [v_1(t), v_2(t), v_3(t)]^{\mathrm{T}}$ describes the ship's motion in different directions, where $v_1(t)$ represents the surge velocity, $v_2(t)$ indicates the sway velocity, and $v_3(t)$ corresponds to the yaw rate. The position vector $\lambda(t) = [x_o(t), y_o(t), \theta(t)]^{\mathrm{T}}$ is used to describe the ship's position

and orientation on the water surface, where $x_o(t)$ and $y_o(t)$ represent the coordinates of the ship in the horizontal plane, and $\theta(t)$ denotes the ship's heading angle. The matrix $\mathcal{C}$ is the damping matrix. The matrix $\mathcal{D}$ represents the mooring moment matrix, which models external disturbances such as wind, waves, and ocean currents acting on the UMVs. The matrix $\mathcal{G}$ is the thrust allocation matrix, responsible for distributing thrust to the ship's propellers. Additionally, the rotation matrix $\mathcal{S}(\theta(t))$ is given by:

$$\mathcal{S}(\theta(t)) = \begin{bmatrix} \cos(\theta(t)) & -\sin(\theta(t)) & 0 \\ \sin(\theta(t)) & \cos(\theta(t)) & 0 \\ 0 & 0 & I \end{bmatrix},$$

For the control of UMVs in the northern region, where the yaw angle $\theta(t)$ is small, the matrix $\mathcal{S}(\theta(t))$ can be approximated by the identity matrix $I$. We define the following matrices $\mathcal{A}_1 = -\xi^{-1}\mathcal{C}$, $\mathcal{B} = \xi^{-1}\mathcal{G}$, and $\mathcal{F} = -\xi^{-1}\mathcal{D}$. let $x(t) = [\lambda^{\mathrm{T}}(t), v^{\mathrm{T}}(t)]^{\mathrm{T}}$. Thus, the dynamic equation of UMVs can be written as follows:

$$\dot{x}(t) = Ax(t) + B_1 u(t) + Fg(t, v(t)) + \varpi(t), \quad (3)$$

where $A = \begin{bmatrix} 0 & I \\ 0 & \mathcal{A}_1 \end{bmatrix}$, $B_1 = \begin{bmatrix} 0 \\ \mathcal{B} \end{bmatrix}$, $F = \begin{bmatrix} 0 \\ \mathcal{F} \end{bmatrix}$. $\varpi(t) \in L_2[0, \infty)$ represents disturbance. Defined reference signal $x_{ref} = \begin{bmatrix} \lambda_{ref} \\ v_{ref} \end{bmatrix}$, the error vector is defined as $e(t) = x(t) - x_{ref}$. The error dynamics of the UMVs can be expressed as follows:

$$\dot{e}(t) = Ae(t) + B_1 u(t) + Fg(t, e(t)) + B_2 \omega(t). \quad (4)$$

let $e(t) \in \mathbb{R}^n$ denote the state vector, $u \in \mathbb{R}^p$ represent the control input vector. The term $B_2 \omega(t)$ is defined as $Ax_{ref} + \varpi(t)$, where $\omega(t) = \begin{bmatrix} x_{ref} \\ \varpi(t) \end{bmatrix}$, and $B_2 = \begin{bmatrix} A & I \end{bmatrix}$. Considering the unavoidable time delay during signal transmission, it follows from equation (4) that:

$$\dot{e}(t) = Ae(t) + A_1 e(t - d) + B_1 u(t) + Fg(e(t), e(t - d)) + B_2 \omega(t), \quad (5)$$

where $d > 0$ represents the time delay, and $g : \mathbb{R}^n \times \mathbb{R}^n \to \mathbb{R}^m$ is assumed to satisfy the following inequality.

*Assumption 1:* Let matrices $\mathbb{N} > 0$ and $\mathbb{Y} > 0$, where $\mathbb{N} \in \mathbb{R}^{m \times m}$ and $\mathbb{Y} \in \mathbb{R}^{2n \times 2n}$. The nonlinear function $g(\cdot)$ satisfies the following inequality:

$$g^{\mathrm{T}}(e(t), e(t - d))\mathbb{N}^{-1} g(e(t), e(t - d))$$
$$\leq [e^{\mathrm{T}}(t) \quad e^{\mathrm{T}}(t - d)]\mathbb{Y}[e^{\mathrm{T}}(t) \quad e^{\mathrm{T}}(t - d)]^{\mathrm{T}}.$$

*Remark 1:* Assumption 1 ensures that the function $g(t)$ is bounded. When $e(t) = 0$ or $e(t - d) = 0$, Assumption 1 in this article is the general form of Assumption 1 in reference [17].

To bring both linear and angular velocities to zero and minimize the impact of external disturbances such as wind,

waves, and currents, the output $\mathcal{Z}(t)$, can be formulated as follows:

$$\mathcal{Z}(t) = C_z e(t) \qquad (6)$$

*Definition 1:* [21] The system is described by

$$\dot{x}(t) = A_d x(t) + B_d \omega(t),$$
$$\mathcal{Z}(t) = C_d x(t), \ x(0) = 0. \qquad (7)$$

Given a constant $\gamma_0 > 0$, $\omega(t) \in L_2[0, \infty)$, if for any $\epsilon > 0$, the following condition

$$\int_0^\infty \mathcal{Z}^{\mathrm{T}}(t)\mathcal{Z}(t)\mathrm{d}t \le \gamma_0^2 \int_0^\infty \omega^{\mathrm{T}}(t)\omega(t)\mathrm{d}t + \epsilon,$$

is satisfied, then the system (7) is said to achieve an adaptive $H_\infty$ performance index that does not exceed $\gamma_0$.

*Definition 2:* The cost function related to system (5) is described as follows:

$$J = \int_0^\infty \left[ e^{\mathrm{T}}(t)\Omega e(t) + u^{\mathrm{T}}(t)\mathbb{R}_q u(t) \right] \mathrm{d}t. \qquad (8)$$

where $\Omega^{\mathrm{T}} = \Omega \ge 0$ and $\mathbb{R}_q^{\mathrm{T}} = \mathbb{R}_q \ge 0$.

A stabilization controller $u(t)$ for system (5) is called a guaranteed cost controller if it ensures that $J \le J^*$, where $J^*$ is a positive scalar. The value $J^*$ is known as the guaranteed cost.

### B. Control Objective

For UMVs (5) affected by time delays, this paper proposes a guaranteed cost DP controller based on the Lyapunov matrix. The controller is designed to drive the state error of the UMVs asymptotically converges to zero, while also satisfying the specified $H_\infty$ performance criteria and guaranteeing an upper limit on the predefined cost function.

### III. PRELIMINARIES

We will construct a complete-type LKF for UMVs (5) based on Lyapunov matrix. In the following section, we begin by defining the Lyapunov matrix.

### A. Lyapunov matrix

We will now present relevant concepts related to linear time-delay systems as follows [22]:

$$\dot{e}(t) = Ae(t) + A_1 e(t - d),$$
$$e(\iota) = \phi(\iota), \ \iota \in [-d, 0], \qquad (9)$$

where $e(t) \in \mathbb{R}^n$ represents the state vector, $d > 0$ is the time delay. $A$, $A_1 \in \mathbb{R}^{n \times n}$ are system matrices.

*Definition 3:* [22] Given a matrix $\mathcal{P} > 0$, if the matrix $Q : [-d, d] \to \mathbb{R}^{n \times n}$ meets the following conditions:

$$\dot{Q}(\pi) = Q(\pi)A + Q(\pi - d)A_1,$$
$$Q(-\pi) = Q^{\mathrm{T}}(\pi),$$
$$-\mathcal{P} = Q(0)A + Q(-d)A_1 + A^{\mathrm{T}}Q(0) + A_1^{\mathrm{T}}Q(d), \qquad (10)$$

*Definition 4:* [22] If the system (9) is asymptotically stable, there exists a Lyapunov matrix $Q(\cdot)$ associated with matrix $\mathcal{P}$ for system (9).

*Lemma 1:* Suppose there are matrices $H = H^{\mathrm{T}} > 0$ and $K_{11} \in \mathbb{R}^{p \times n}$, and for any $U > 0$, the following LMI condition is satisfied:

$$\begin{bmatrix} \Lambda_2 & A_1 X \\ (A_1 X)^{\mathrm{T}} & -U \end{bmatrix} < 0 \qquad (11)$$

where $\Lambda_2 = AX - B_1 Y_1 + (AX - B_1 Y_1)^{\mathrm{T}} + U$, $X = H^{-1}$, $Y_1 = K_{11}H^{-1}$, and $U = H^{-1}LH^{-1}$, then there exists a controller $u_1(t) = -K_{11}e(t)$ that guarantees system (9) is asymptotically stable.

*Proof 1:* Select the Lyapunov function:

$$V_c(e(t)) = e^{\mathrm{T}}(t)He(t) + \int_{t-d}^t e^{\mathrm{T}}(\theta)Le(\theta)\mathrm{d}\theta.$$

We can derive:

$$\left. \frac{\mathrm{d}V_c(e(t))}{\mathrm{d}t} \right|_{(9)} = \Lambda_0^{\mathrm{T}}\Omega_1\Lambda_0$$

where

$$\Lambda_0 = [e^{\mathrm{T}}(t), e^{\mathrm{T}}(t-d)]^{\mathrm{T}},$$
$$\Omega_1 = \begin{bmatrix} \Lambda_2 & A_1 X \\ (A_1 X)^{\mathrm{T}} & -U \end{bmatrix},$$
$$\Lambda_2 = AX - B_1 Y_1 + (AX - B_1 Y_1)^{\mathrm{T}} + U,$$
$$X = H^{-1}, \ Y_1 = K_{11}H^{-1}, \ U = H^{-1}LH^{-1}.$$

Using Lyapunov stability theory, the controller $u_1(t) = -K_{11}e(t)$ guarantees the asymptotic stability of system (9).

### IV. A COMPLETE–TYPE LKF

We construct a LKF $\mathfrak{V}(\cdot)$:

$$\mathfrak{V}(e(t)) = \mathfrak{V}_1(e(t)) + \mathfrak{V}_2(e(t)), \ e \in C_p([-d, 0], \mathbb{R}^n), \quad (12)$$

where

$$\mathfrak{V}_1(e(t)) = e^{\mathrm{T}}(t)Q(0)e(t) + 2e^{\mathrm{T}}(t)\Gamma_1(e(t))$$
$$+ \int_{-d}^0 \int_{-d}^0 e^{\mathrm{T}}(t + \tau_1)A_1^{\mathrm{T}}Q(\tau_1 - \tau_2)A_1 e(t + \tau_2)\mathrm{d}\tau_1\mathrm{d}\tau_2,$$
$$\mathfrak{V}_2(e(t)) = \int_{-d}^0 \int_\tau^0 e^{\mathrm{T}}(t + s)A_1^{\mathrm{T}}Q^{\mathrm{T}}(-d - \tau)\mathcal{R}Q(-d - \tau)$$
$$\times A_1 e(t + s)\mathrm{d}s\mathrm{d}\tau + \int_{-d}^0 e^{\mathrm{T}}(t + \tau)\mathcal{Q}_1 e(t + \tau)\mathrm{d}\tau, \qquad (13)$$

where $\Gamma_1(e(t)) = \int_{-d}^0 Q(-d - \tau)A_1 e(t + \tau)\mathrm{d}\tau$ and matrices $\mathcal{R}$, $\mathcal{Q}_1$ satisfying the $\mathcal{R}^{\mathrm{T}} = \mathcal{R} > 0$, $\mathcal{Q}_1^{\mathrm{T}} = \mathcal{Q}_1 > 0$.

### V. CONTROLLER DESIGN AND STABILITY ANALYSIS

In this section, we will provide a detailed explanation of the controller design process and conduct a systematic analysis of its stability.

## A. Controller Design

We propose the following guaranteed cost DP controller for UMVs in (5):

$$u(t) = u_1(t) + u_2(t),$$
$$u_1(t) = -K_{11}e(t),$$
$$u_2(t) = \frac{1}{2}K_{21}B_1^{\mathrm{T}}[Q(0)e(t) + \Gamma_1(e(t))] + \frac{1}{2}K_{22}e(t-d),$$

(14)

where $K_{11}$, $K_{21}$, $K_{22}$ are feedback gain matrices. $K_{11}$ is already determined in Lemma 1, while $K_{21}$ and $K_{22}$ will be provided in Theorem 1.

*Theorem 1:* Consider the UMVs (5) under Assumption 1. The guaranteed cost DP controller is defined by (14). For the given positive definite matrices $\mathbb{N} \in \mathbb{R}^{m \times m}$, $\mathbb{Y} := \begin{bmatrix} \mathbb{Y}_{11} & \mathbb{Y}_{12} \\ \mathbb{Y}_{12}^{\mathrm{T}} & \mathbb{Y}_{22} \end{bmatrix} \in \mathbb{R}^{2n \times 2n}$, $\mathcal{P} \in \mathbb{R}^{n \times n}$, and a positive constant $\gamma_0$, if there exist positive definite matrices $\mathcal{R}, \mathcal{Q}_1 \in \mathbb{R}^{n \times n}$, and matrices $K_{21} \in \mathbb{R}^{p \times p}, K_{22} \in \mathbb{R}^{p \times n}$ such that $\mathcal{P} - \mathcal{Q}_1 - \mathcal{P}_1 > 0$ and the following inequality holds,

$$E := \begin{bmatrix} \mathcal{P} + \mathcal{Q}_1 + \mathcal{P}_1 - E_1 & E_2 & E_3 \\ E_2^{\mathrm{T}} & -\mathcal{Q}_1 + \mathbb{Y}_{22} & \frac{1}{2}K_{22}^{\mathrm{T}}B_1^{\mathrm{T}} \\ E_3^{\mathrm{T}} & \frac{1}{2}B_1K_{22} & E_4 \end{bmatrix} < 0,$$

(15)

where

$$E_1 = \frac{1}{2}Q(0)B_1(K_{21} + K_{21}^{\mathrm{T}})B_1^{\mathrm{T}}Q(0) - \mathbb{Y}_{11} - C_z^{\mathrm{T}}C_z$$
$$\quad - \gamma_0^{-2}Q(0)B_2B_2^{\mathrm{T}}Q(0) - Q(0)F\mathbb{N}F^{\mathrm{T}}Q(0),$$
$$E_2 = \frac{1}{2}Q(0)B_1K_{22} + \mathbb{Y}_{12},$$
$$E_3 = Q(0)B_1K_{21}B_1^{\mathrm{T}} + Q(0)F\mathbb{N}F^{\mathrm{T}} + \gamma_0^{-2}Q(0)B_2B_2^{\mathrm{T}},$$
$$E_4 = -\frac{\mathcal{R}}{d} + B_1K_{21}B_1^{\mathrm{T}} + F\mathbb{N}F^{\mathrm{T}} + \gamma_0^{-2}B_2B_2^{\mathrm{T}},$$

then, the state of the UMVs in system (5) asymptotically converge to zero, while maintaining an $H_\infty$ norm bound of $\gamma_0$.

*Proof 2:* The time derivative of $\mathfrak{V}(e(t))$ along the trajectory of the UMVs (5) can be calculated as follows:

$$\left.\frac{\mathrm{d}\mathfrak{V}(e(t))}{\mathrm{d}t}\right|_{(5)} + \mathcal{Z}^{\mathrm{T}}(t)\mathcal{Z}(t) - \gamma_0^2\omega^{\mathrm{T}}(t)\omega(t)$$
$$= -U_0(e(t)) + \mathcal{Z}^{\mathrm{T}}(t)\mathcal{Z}(t) - \gamma_0^2\omega^{\mathrm{T}}(t)\omega(t)$$
$$\quad + 2g^{\mathrm{T}}(e(t), e(t-d))F^{\mathrm{T}}[Q(0)e(t) + \Gamma_1(e(t))]$$
$$\quad + 2[Q(0)e(t) + \Gamma_1(e(t))]^{\mathrm{T}}B_2\omega(t)$$
$$\quad + 2[Q(0)e(t) + \Gamma_1(e(t))]^{\mathrm{T}}B_1u(t)$$

(16)

where

$$U_0(e) = e^{\mathrm{T}}(t)(\mathcal{P} - \mathcal{Q}_1 - \mathcal{P}_1)e(t) + e^{\mathrm{T}}(t-d)\mathcal{Q}_1e(t-d)$$
$$\quad + \int_{-d}^0 e^{\mathrm{T}}(t+\tau)A_1^{\mathrm{T}}Q^{\mathrm{T}}(-d-\tau)\mathcal{R}Q(-d-\tau)A_1e(t+\tau)\mathrm{d}\tau.$$
$$\mathcal{P}_1 = \int_{-d}^0 A_1^{\mathrm{T}}Q^{\mathrm{T}}(-d-\tau)\mathcal{R}Q(-d-\tau)A_1\mathrm{d}\tau.$$

Substituting (14) into (16), we have

$$\left.\frac{\mathrm{d}\mathfrak{V}(e(t))}{\mathrm{d}t}\right|_{(5)} + \mathcal{Z}^{\mathrm{T}}(t)\mathcal{Z}(t) - \gamma_0^2\omega^{\mathrm{T}}(t)\omega(t) \le \Gamma^{\mathrm{T}}(t)E\Gamma(t)$$

(17)

where

$$\Gamma(t) = [e^{\mathrm{T}}(t)\ e^{\mathrm{T}}(t-d)\ \Gamma_1^{\mathrm{T}}(e(t))]^{\mathrm{T}},$$

$$E := \begin{bmatrix} \mathcal{P} + \mathcal{Q}_1 + \mathcal{P}_1 - E_1 & E_2 & E_3 \\ E_2^{\mathrm{T}} & -\mathcal{Q}_1 + \mathbb{Y}_{22} & \frac{1}{2}K_{22}^{\mathrm{T}}B_1^{\mathrm{T}} \\ E_3^{\mathrm{T}} & \frac{1}{2}B_1K_{22} & E_4 \end{bmatrix},$$

where

$$E_1 = \frac{1}{2}Q(0)B_1(K_{21} + K_{21}^{\mathrm{T}})B_1^{\mathrm{T}}Q(0) - \mathbb{Y}_{11} - C_z^{\mathrm{T}}C_z$$
$$\quad - \gamma_0^{-2}Q(0)B_2B_2^{\mathrm{T}}Q(0) - Q(0)F\mathbb{N}F^{\mathrm{T}}Q(0),$$
$$E_2 = \frac{1}{2}Q(0)B_1K_{22} + \mathbb{Y}_{12},$$
$$E_3 = Q(0)B_1K_{21}B_1^{\mathrm{T}} + Q(0)F\mathbb{N}F^{\mathrm{T}} + \gamma_0^{-2}Q(0)B_2B_2^{\mathrm{T}},$$
$$E_4 = -\frac{\mathcal{R}}{d} + B_1K_{21}B_1^{\mathrm{T}} + F\mathbb{N}F^{\mathrm{T}} + \gamma_0^{-2}B_2B_2^{\mathrm{T}},$$

For $E < 0$, it implies that

$$\left.\frac{\mathrm{d}\mathfrak{V}(e(t))}{\mathrm{d}t}\right|_{(5)} + \mathcal{Z}^{\mathrm{T}}(t)\mathcal{Z}(t) - \gamma_0^2\omega^{\mathrm{T}}(t)\omega(t) \le 0.$$

(18)

If Theorem 1 holds, then the $\int_{t_0}^t \Gamma^{\mathrm{T}}(\tau)E\Gamma(\tau)\mathrm{d}\tau < 0$ is satisfied:

$$0 \le \epsilon_{min}\|e(t)\|^2 \le \mathfrak{V}(e) \le \mathfrak{V}(e(t_0)) - \int_{t_0}^t \mathcal{Z}^{\mathrm{T}}(\tau)\mathcal{Z}(\tau)\mathrm{d}\tau$$
$$+ \gamma_0^2 \int_{t_0}^t \omega^{\mathrm{T}}(\tau)\omega(\tau)\mathrm{d}\tau, \ t > t_0.$$

(19)

Clearly

$$\lim_{t \to \infty} \int_{t_0}^t \Gamma^{\mathrm{T}}(\tau)E\Gamma(\tau)\mathrm{d}\tau \le \mathfrak{V}(e(t_0)).$$

(20)

We obtain

$$\lim_{t \to \infty}\|e(t)\| = 0,$$

(21)

By integrating equation (18)) from 0 to $\infty$, we obtain

$$\int_0^\infty \mathcal{Z}^{\mathrm{T}}(t)\mathcal{Z}(t)\mathrm{d}t \le \gamma_0^2 \int_0^\infty \omega^{\mathrm{T}}(t)\omega(t)\mathrm{d}t + \mathfrak{V}(0).$$

(22)

## B. Guaranteed Cost Analysis

When the disturbance $\omega(t)$ is absent, combining (8), (14), and (18) yields:

$$\left.\frac{d\mathfrak{V}(e(t))}{dt}\right|_{(5)} + e^{\mathrm{T}}(t)\Omega e(t) + u^{\mathrm{T}}(t)\mathbb{R}_q u(t)$$
$$\le \Gamma^{\mathrm{T}}(t)\left(E + \mathrm{diag}(\Omega, 0, 0) + \frac{1}{4}O^{\mathrm{T}}\mathbb{R}_q O\right)\Gamma(t)$$

(23)

where

$$O = \begin{bmatrix} -(\mathbb{Y} + K_{21})B_1^{\mathrm{T}}Q(0) & K_{22} & -(\mathbb{Y} + K_{21})B_1^{\mathrm{T}} \end{bmatrix}.$$

We have

$$\begin{bmatrix} E + \mathrm{diag}(\Omega, 0, 0) & O^{\mathrm{T}} \\ O & -4\mathbb{R}_q^{-1} \end{bmatrix} < 0$$

Hence,

$$\int_0^\infty \left[ e^{\mathrm{T}}(t)\Omega e(t) + u^{\mathrm{T}}(t)\mathbb{R}_q u(t) \right] \mathrm{d}t \le J^*.$$

where $J^* = \mathfrak{V}(e(t))$, with $\mathfrak{V}(e(t))$ defined in (12).

## VI. SIMULATION EXAMPLE

The proposed control method's effectiveness is demonstrated through a standard floating production vessel model, as referenced in [23]. The matrices $\xi$, $\mathcal{C}$, and $\mathcal{D}$ are specified in [23], and the thruster configuration matrix $\mathcal{G}$ is derived from [24].

The initial condition is given as $\phi(s) = [0\ 0\ 0\ 0\ 0\ 0.2]^{\mathrm{T}}$, with the reference signal set to $x_{ref} = [0.01\ -0.01\ 0.05\ 0.01\ 0.04\ 0.01]^{\mathrm{T}}$. The time delay is $d = 1$, and the $H_\infty$ performance index $\gamma_0 = 2$.

The controller gain matrix $K_{11}$ is obtained by solving the LMI (11) from Lemma 1, as follows:

$$K_{11} = \begin{bmatrix} 3.7401 & -1.0550 & 1.6703 \\ 3.5625 & -0.3782 & 0.8900 \\ -1.8457 & 7.7381 & -7.8852 \\ -1.7986 & 7.5585 & -7.6782 \\ -0.2156 & 1.5274 & -0.7243 \\ -0.4379 & 2.3744 & -1.7009 \end{bmatrix}$$

$$\begin{bmatrix} 3.8794 & -0.4071 & 0.6533 \\ 3.8305 & 0.0888 & 0.2145 \\ -0.4836 & 5.9344 & -4.2105 \\ -0.4706 & 5.8028 & -4.0941 \\ -0.0351 & 1.3831 & -0.1833 \\ -0.0963 & 2.0038 & -0.7325 \end{bmatrix}.$$

We set the matrix $\mathcal{Q} = I$. The $(i, j)$-th element of the matrix $Q(\theta)$, denoted as $Q_{ij}(\theta)$, is determined using the method proposed in [22]. Figures 1-2 show the values of $Q_{ij}(\theta)$ for $\theta \in [0, 1]$.

Finally, by solving LMI (15) as described in Theorem 1, the controller gain matrices $K_{21}$ and $K_{22}$ are computed as:

$$K_{21} = 1 \times 10^4 \begin{bmatrix} 0.0284 & 0.0561 & 0.0446 \\ -0.0249 & -0.0535 & -0.0615 \\ -0.0160 & 0.0215 & 0.0366 \\ 0.0187 & -0.0010 & -0.0542 \\ -0.2113 & 0.2496 & -0.1101 \\ -0.0871 & 0.0356 & -0.0328 \end{bmatrix}$$

$$\begin{bmatrix} 0.0381 & -0.0108 & -0.0257 \\ -0.0140 & 0.0119 & 0.0273 \\ 0.0723 & -0.0709 & -0.0315 \\ -0.0511 & 0.0035 & 0.1249 \\ -0.0808 & -0.9459 & 1.2040 \\ 0.1207 & 0.5283 & -0.6940 \end{bmatrix}.$$

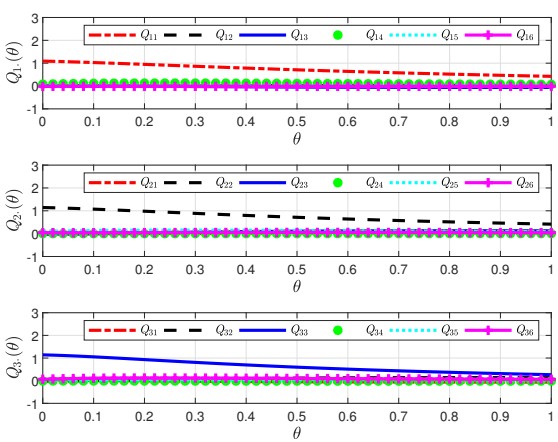

Figure 1. Lyapunov matrix $Q_{ij}(\theta)$, (i=1,2,3 j = 1,2,3,4,5,6).

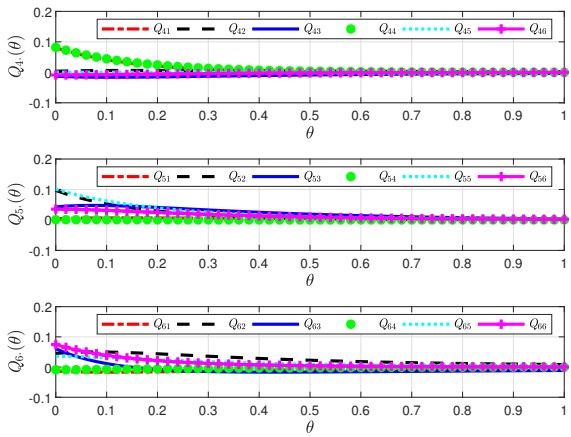

Figure 2. Lyapunov matrix $Q_{ij}(\theta)$, (i=4,5,6 j = 1,2,3,4,5,6).

$$K_{22} = \begin{bmatrix} -15.4416 & 8.9036 & 66.6063 \\ -18.9989 & -43.3441 & -101.0469 \\ 22.5784 & 53.9648 & 21.5477 \\ -82.2859 & -141.7415 & 16.5537 \\ -118.7051 & -303.3277 & 414.2256 \\ 118.3731 & 331.0541 & -512.3389 \end{bmatrix}$$

$$\begin{bmatrix} 35.6011 & 15.1773 & -22.0347 \\ -70.0417 & -49.6179 & -12.4059 \\ -26.9017 & 10.6883 & 43.2947 \\ -69.8399 & -34.0587 & -92.0165 \\ -396.6715 & 76.7366 & -41.8016 \\ 433.3628 & -113.3949 & 30.4866 \end{bmatrix}.$$

Figures 3-4 illustrate the trajectories of the position error, yaw angle error, and velocity error for UMVs (5). Figure 5 shows the control inputs produced by the controller as defined in (14).

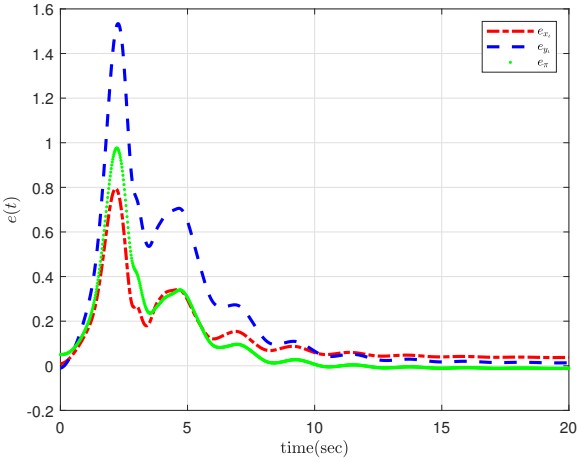

Figure 3. Response curves of UMVs position and yaw angle error.

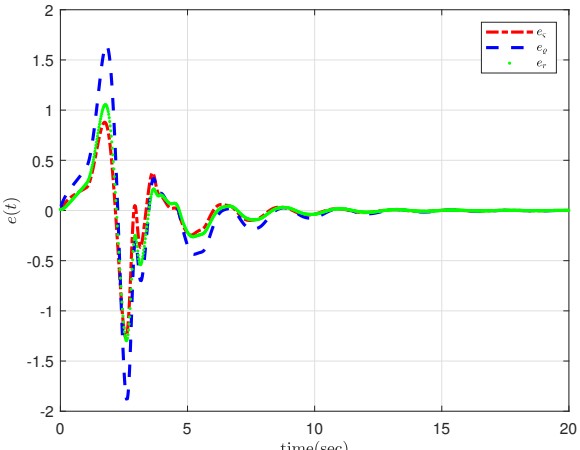

Figure 4. Response curves of UMVs velocity error.

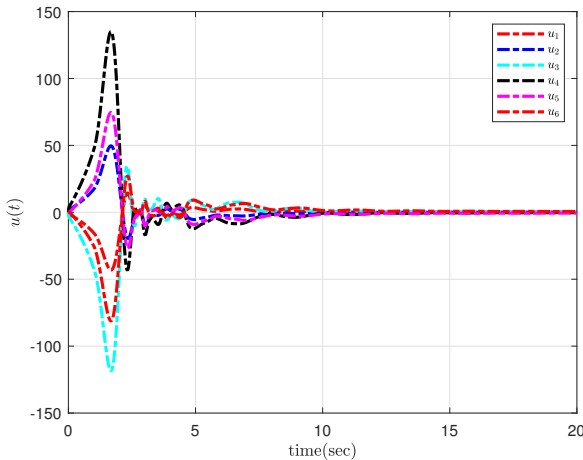

Figure 5. The comparison of response curves for $u(t)$

In Figure 3, it is clear that the error curves under the proposed control initially exhibit small fluctuations before gradually converging to zero. This demonstrates the effectiveness of the proposed control strategy. Figure 5 illustrates the response curves of the guaranteed cost DP controller $u(t)$.

## CONCLUSION

In this paper, we have addressed the guaranteed cost dynamic positioning control problem for UMVs with time delays. First, we propose a complete-type LKF for UMVs with time delays, which leads to less conservativeness. Furthermore, a novel approach for designing guaranteed cost dynamic positioning controller for DP systems is proposed. The specific form of this controller is derived from feasible solutions of LMIs. The proposed method was validated through simulation, demonstrating its effectiveness. Future work will focus on extending the control strategy to systems with time-varying delays, further enhancing the robustness of DP control for UMVs.

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
