# OpenReview forum: "Lyapunov Matrix-Based Guaranteed Cost Dynamic Positioning Control for Unmanned Marine Vehicles With Time Delay"
_IEEE.org/ICIST/2024/Conference — IEEE ICIST 2024 Conference Submission_

### Official Review · Reviewer_STzX · 2024-08-21
**Accept**

**Rating:** 8
**Confidence:** 5

**Review:**

The paper presents a well-structured and rigorous approach to designing a guaranteed cost dynamic positioning controller for UMVs under time delays.  The proposed method is based on solid theoretical foundations, utilizing the LKF framework, LMI technique, and Jensen's inequality to derive sufficient conditions for the controller's existence.  The simulation results demonstrate the effectiveness of the approach, providing evidence of its practical value.The following comments should be considered to improve the paper’s quality further.1.Compared to conventional time delays compensation techniques, what are the specific advantages of the time delays compensation method proposed in this paper?2. In guaranteed cost analysis, the disturbance is considered to be absent, what is the purpose of this process?

---

### Official Review · Reviewer_xFUR · 2024-08-24
**The paper is written clearly, exceptionally excellent.**

**Rating:** 8
**Confidence:** 3

**Review:**

This paper excels in terms of quality, clarity, originality, and significance, but I would still like to offer some suggestions.
1. Can the method proposed in this paper be applied to actual systems?
2. Discuss the rationality of the assumptions made in the paper.

---

### Official Review · Reviewer_Zr5v · 2024-08-25
**new insight and opinion into the development of  UMVs with time delays**

**Rating:** 8
**Confidence:** 4

**Review:**

In this manuscript  titled "Lyapunov Matrix-Based Guaranteed Cost Dynamic Positioning Control for Unmanned Marine Vehicles With Time Delay", the authors summarized and discussed a Lyapunov matrix-based dynamic positioning controller for unmanned marine vehicles (UMVs) that accounts for time delays using a novel Lyapunov–Krasovskii functional and ensures state errors converge to zero with bounded performance through linear matrix inequalities and Jensen’s inequality.This work provides new insight and opinion into the development of  UMVs with time delays,a novel approach for designing guaranteed cost dynamic positioning controller for DP systems is proposed. The manuscript is well-organized and clearly stated. I would suggest accepting it after the following concerns are addressed: Analysis and complete interpretation of all data

---

### Decision · Program_Chairs · 2024-09-08

Accept (Oral)